# Biomolecular analyses reveal the age, sex and species identity of a near-intact Pleistocene bird carcass

Nicolas Dussex [1,2,3], David W.G. Stanton[1,2], Hanna Sigeman [4], Per G.P. Ericson[2], Jacquelyn Gill[5,6], Daniel C. Fisher[7], Albert V. Protopopov[8], Victoria L. Herridge[9], Valery Plotnikov [8], Bengt Hansson [4] & Love Dalén[1,2,3 ✉]

Ancient remains found in permafrost represent a rare opportunity to study past ecosystems. Here, we present an exceptionally well-preserved ancient bird carcass found in the Siberian permafrost, along with a radiocarbon date and a reconstruction of its complete mitochondrial genome. The carcass was radiocarbon dated to approximately 44–49 ka BP, and was genetically identified as a female horned lark. This is a species that usually inhabits open habitat, such as the steppe environment that existed in Siberia at the time. This near-intact carcass highlights the potential of permafrost remains for evolutionary studies that combine both morphology and ancient nucleic acids.

[1] Centre for Palaeogenetics, Svante Arrhenius väg 20C, SE-106 91 Stockholm, Sweden. [2] Department of Bioinformatics and Genetics, Swedish Museum of Natural History, Box 50007, SE-10405 Stockholm, Sweden. [3] Department of Zoology, Stockholm University, SE-10691 Stockholm, Sweden. [4] Department of Biology, Lund University, Ecology Building, 223 62 Lund, Sweden. [5] School of Biology & Ecology, 134 Sawyer Research Labs, University of Maine, ME 04469 Orono, USA. [6] Climate Change Institute, 134 Sawyer Research Labs, University of Maine, ME 04469 Orono, USA. [7] Museum of Paleontology, Department of Earth and Environmental Sciences, University of Michigan, 1100 North University Avenue, MI 48109-1079 Ann Arbor, USA. [8] Department of Study of Mammoth Fauna, The Academy of Science of the Sakha Republic, 677027 Lenin Avenue 33, Yakutsk, Republic of Sakha (Yakutia), Russia. [9] Department of Earth Sciences, Natural History Museum, Cromwell Road, SW7 5BD London, UK. ✉email: love.dalen@nrm.se

Permafrost deposits containing both animal and plant material represent a unique opportunity to reconstruct paleoenvironments[1,2]. In recent years, permafrost sites in the Arctic have revealed a wealth of frozen animal carcasses from the last glaciation, including mammoths, woolly rhinoceroses, horses, bisons, and wolverines[3,4]. These remains are of great interest to paleontology since they enable a better understanding of the impact of climate change on species, populations, and communities[1,2]. Moreover, because such frozen carcasses are often extremely well preserved, they allow for studies of morphological traits, as well as the ecology and evolution of a range of extinct and extant animal species (e.g.[5–7]). The high degree of preservation of both tissue and DNA in permafrost remains has been particularly beneficial to the field of ancient DNA (aDNA). For instance, sequencing of DNA from Pleistocene and Holocene mammal remains has provided a better understanding of the behaviour of extinct species[5,8], while the sequencing of complete Pleistocene genomes has allowed studies of evolutionary rates, extinction dynamics and inter-specific hybridisation[9–11]. However, most examples of this type of frozen permafrost tissues have been from large mammals. To our knowledge, no frozen bird carcasses have yet been described from late Pleistocene permafrost deposits.

Here, we present a complete bird specimen located within an undisturbed permafrost sequence in the Belaya Gora area of north-eastern Siberia. We radiocarbon dated this specimen and used high-throughput shotgun sequencing to identify the species through reconstruction of its complete mitogenome. This unique find highlights the importance of preserving newly discovered permafrost remains for future genomic and morphometric studies of Pleistocene fauna.

## Results

### Radiocarbon dating and DNA sequencing of bird carcass found in permafrost.

The frozen carcass of a near-complete passerine (Fig. 1) was recovered in permafrost from a site 30 km east from the village of Belaya Gora, Yakutia (N68.57887, E147.16055; Fig. 2). The site comprises a series of tunnels that have been hydraulically mined into the permafrost by fossil ivory hunters, and is located on the small Tirekhtyakh River, which is a tributary to the Indigirka River. The bird carcass was found ~150 lateral meters into one of the tunnels, at a depth of roughly 7 meters from the surface.

The bird was radiocarbon dated to 42,600 +/− 1100 years BP (OxA-38572), which corresponds to a calibrated age between 44,163–48,752 Cal years BP[12]. We subsequently used ~50 mg of the bird tissue for DNA extraction and whole genome shotgun-sequencing. To identify the species, we then mapped the shotgun-sequencing data to the chicken (Gallus gallus; GenBank:

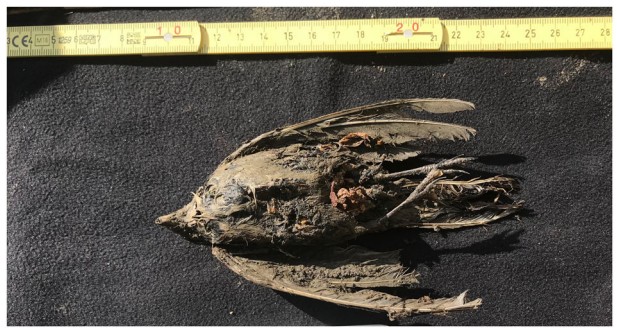

**Fig. 1 Ancient bird carcass.** The photo shows the ventral view of the c. 46 ka old bird carcass recovered from the Siberian permafrost (Photo: Love Dalén).

NC_001323) mitogenome. Out of a total of 85,408,658 reads, 3,663 (<0.01%) mapped against the chicken mitogenome. While the reconstructed mitogenome was incomplete, we extracted a partial COI gene (232 bp), which is routinely used for species identification. We then searched for matches with this gene in GenBank avian genetic databases using BLASTn[13,14] and found a 100% identity match with the horned lark (Eremophila alpestris).

To estimate the overall endogenous DNA content of the ancient horned lark extract, we then mapped the shotgun-sequencing data to a nuclear genome assembly from a modern horned lark (Sigeman et al.[15]; N50 = 20 kb; subspecies E. a. flava). Out of 8,5408,658 reads, 757,883 (0.89%) reads mapped to this nuclear genome. The average genome depth was of 0.032X (min = 0; max = 116), with 2.5% of the assembly having a coverage ≥1X. We also found that there were on average 37 and 78 reads per 100 kb mapping to Z-linked and autosomal contigs, respectively. This corresponds to a ratio of Z-linked/autosomal reads of 0.47. Since female birds are the heterogametic sex (ZW), and since the female-specific W chromosome is highly degraded and differentiated to the Z chromosome over most of its length[15], this ratio indicates that the specimen was a female.

Next, to improve the quality of the ancient horned lark mitogenome, we remapped the shotgun-sequencing data against a modern horned lark de novo mitogenome[15], which we generated using Mitobim v1.9.0[16]. This resulted in an ancient consensus mitogenome with an average depth of 134X (min = 18; max = 264) and 40,685 (0.05%) reads mapping to the modern horned lark mitogenome. To confirm species identity, we then extracted the complete COI (683 bp) and cytb (987 bp) genes, which had a 99% and 95% identity match with horned lark, respectively.

### Phylogenetic placement of the ancient bird specimen.

A comprehensive study on lark phylogeny resolving the complex relationships among larks (Alaudidae) indicated dramatic morphological divergence in some lineages as well as multiple examples of parallel morphological evolution[17]. Within larks, the systematics and evolution of the genus Eremophila were until recently poorly understood, with many subspecies described (owing in part to the variation and divergence in plumage[18,19]). However, a recent study from Ghorbani et al.[20] based on ND2 and cytb genes identified four distinct Eremophila lineages that diverged in the late Pliocene to early Pleistocene. Furthermore, E. alpestris may have originated in north-eastern Siberia during the middle Pleistocene[18,20], before diverging into Eurasian and North American clades.

Specimens such as this ancient lark present an opportunity to understand the evolution and biogeography of this poorly resolved Pleistocene species. We thus examined the phylogenetic placement of the ancient horned lark within the diversity of extant larks using the modern ND2 and cytb (2034 bp) data from Ghorbani et al.[20] and a Bayesian relaxed clock model implemented in BEAST2[21]. The ancient horned lark was placed close to the node between E. a. flava and E. a. brandti (Fig. 3), which are nowadays distributed across the north Palearctic (Scandinavia, Northern Russia) and central Palearctic aridlands (Kazakhstan, Mongolia, China), respectively[19,22]. Pollen records from the Yakutia region indicate that the habitat was dominated by a mix of tundra and steppe environments between 50 and 30 ka BP[23], making E. a. flava the most likely ecological analogue to this ancient specimen[18]. Notably there was low posterior branch support (<0.5) for assigning the ancient specimen to either of these subspecies. Based on this, we hypothesize that the ancient specimen may have belonged to an ancestral population for E. a. flava and E. a. brandti, and that these two lineages subsequently evolved into separate subspecies as a consequence of environmental changes during the Pleistocene/

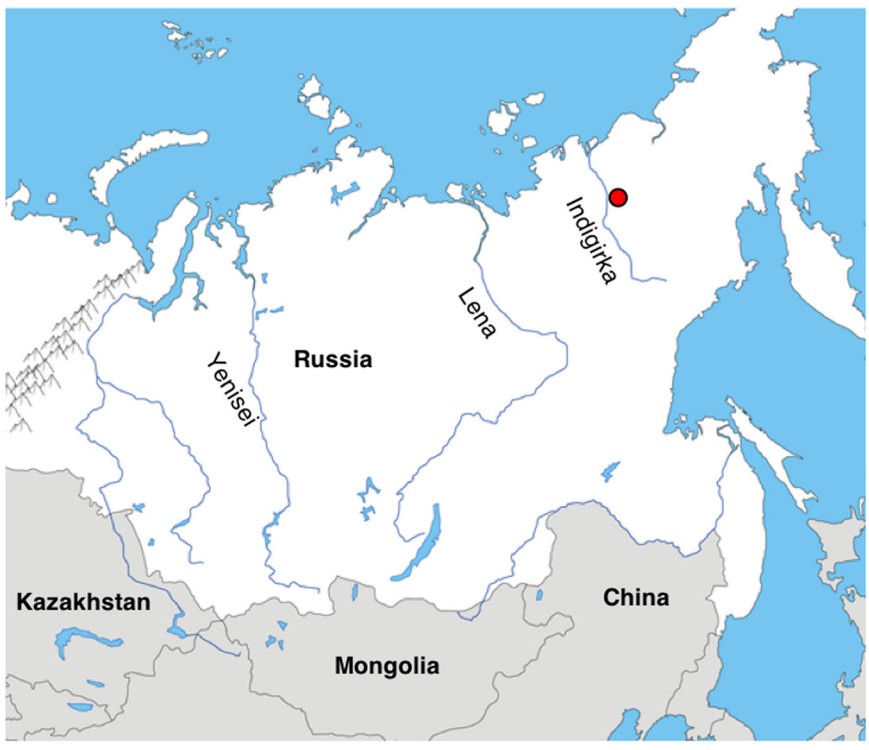

**Fig. 2 Sampling collection site in north-eastern Siberia.** The red dot indicates the sampling location of the ancient bird carcass.

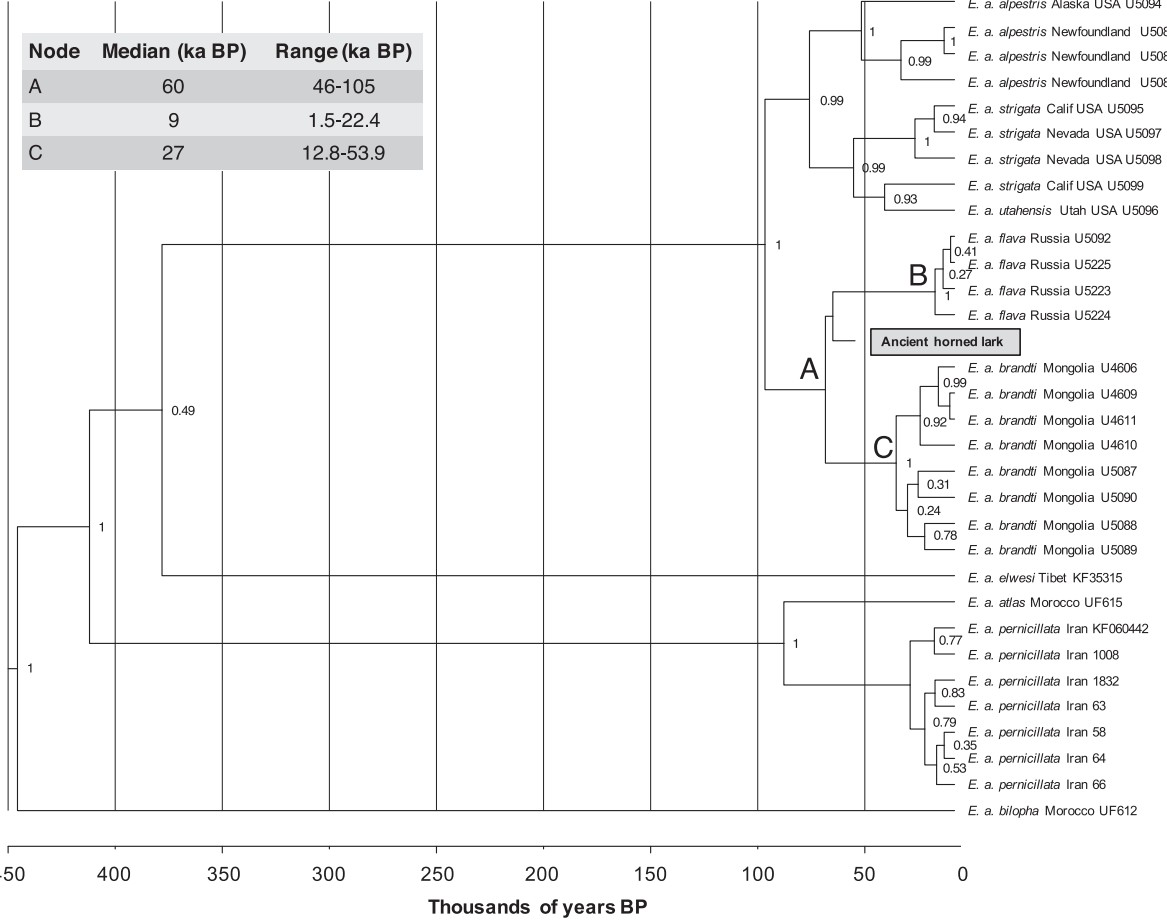

**Fig. 3 Phylogenetic placement of the ancient horned lark.** The phylogeny is inferred from the Bayesian analysis of concatenated ND2 and cyt*b* (2034 bp) for 31 horned lark from Ghorbani et al.[20]. The ancient horned lark highlighted in gray. Nodes with <0.5 support should be considered collapsed.

Holocene transition. Such a Holocene divergence into tundra and steppe forms is consistent with earlier work by Edwards et al.[24] proposing a general stratification from a mixed steppe-tundra into separate tundra and steppe biomes at the end of the last glaciation.

## Discussion

The methods used by fossil ivory hunters may cause damage to animal remains of special scientific value, particularly those with smaller body size that are not easily identified during hydraulic mining. Nevertheless, the exceptionally high degree of morphological preservation of this specimen and of similarly well-preserved animal remains from the area around Belaya Gora holds great promise for further research into the region's evolutionary history. For instance, molecular identification of the sex of animal remains, which is not always possible based on skeletal remains, allows for investigations into the behavioural ecology of extinct species[5,8], while sequencing of mitochondrial or nuclear genomes can enable studies of temporal range shifts associated with past climatic fluctuations[25], as well as reconstructing the past demography in extant and extinct species[11,26,27]. Moreover, because estimates of substitution rates from modern pedigree data can be biased[28], whole genome sequencing of permafrost specimens could provide more accurate estimation of molecular clocks and thus allow improved studies of the micro-evolution of a range of species (e.g., [29,30]). Finally, one especially important advantage of frozen carcasses lies in the preservation of several different tissues and organs. Sequencing RNA from these tissues would give information beyond what is achievable from genomic data alone, enabling studies of gene expression, as shown recently in a study on ancient wolf remains dating back some 14 ka BP[31]. Pleistocene tissue remains recovered from permafrost thus have the potential to become instrumental in better understanding processes such as biological regulation and gene expression in relation to past climate change.

## Methods

**Sample collection and radiocarbon dating**. Following retrieval of the near-complete bird carcass (Figs. 1 and 4) from the permafrost tunnel located along the Tirekhtyakh River (Fig. 2), we excised ~120 mg of feather from the specimen and

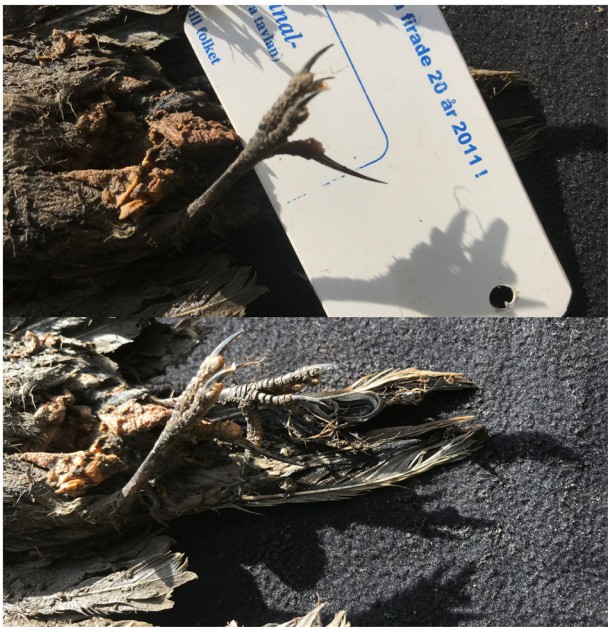

**Fig. 4 Foot of the ancient bird carcass.** Close-up of the foot of the the c. 46 ka old bird carcass recovered from the Siberian permafrost (Photo: Love Dalén).

used this for radiocarbon dating at the Oxford Radiocarbon Accelerator Unit (ORAU) using the Oxcal computer program (v4.3)[12]. In addition, a small piece of tissue (~300 mg) was collected and stored at −20 °C for subsequent DNA analysis. The carcass itself has been accessioned into the collections at the Sakha Academy of Sciences in Yakutsk (accession ID: 2018-Tir-Shorelark-01).

**DNA extraction and sequencing**. After an overnight digestion of ~50 mg of bird tissue in a buffer optimised to digest keratin-rich tissues[32], we extracted DNA using the DNeasy Blood & Tissue Kit (Qiagen, Hilden, Germany). We then built a double-stranded Illumina library according to Meyer & Kircher[33]. We used 20 µl of DNA extract in a 40 µl blunt-end repair reaction with the following final concentration: 1×buffer Tango, 100 µM of each dNTP, 1 mM ATP, 20 U T4 polynucleotide kinase (Thermo Scientific), and 3U USER enzyme (New England Biolabs). Treatment with USER enzyme was performed to excise uracil residues resulting from post-mortem damage[34,35]. The sample was incubated for 3 h at 37 °C, followed by the addition of 0.8 µl T4 DNA polymerase (Thermo Scientific), and incubation at 25 °C for 15 min and 12 °C for 5 min and was then cleaned using MinElute (Qiagen, Hilden, Germany) spin columns following the manufacturer's protocol and eluted in 20 µl EB Buffer. Next, we performed an adapter ligation step where DNA fragments of the library were ligated to a combination of incomplete, partially double-stranded P5- and P7-adapters. This reaction was performed in a 40 µl reaction volume using 20 µl of blunted DNA from the clean-up step and 2 µl P5–P7 adapter mix (0.5 µM final concentration for each adapter) per sample with a final concentration of 1×T4 DNA ligase buffer, 5% PEG-4000, 5U T4 DNA ligase (Thermo Scientific). The sample was incubated for 30 min at room temperature and cleaned using MinElute spin columns as described above. Next, we performed an adapter fill-in reaction in 40 µl final volume using 20 µl adapter-ligated DNA with a final concentration of 1× Thermopol Reaction Buffer, 250 µM of each dNTP, 12U *Bst* Polymerase (Thermo Scientific), Long Fragments. The library was incubated at 37 °C for 20 min and heat-inactivated at 80 °C for 20 min.

This library was then used as stock for indexing PCR amplification using one indexing PCR amplification with double-indexed P5–P7 primers. The amplification was performed in 25 µl volumes with 3 µl of adapter-ligated library as template, with the following final concentrations: 1x AccuPrime reaction mix, 0.3 µM P7-P5 indexing primer mix, 7 U AccuPrime Pfx (Thermo Scientific), and the following cycling protocol: 95 °C for 2 min, 12 cycles at 95 °C for 30 s, 55 °C for 30 s, 72 °C for 1 min, and a final extension at 72 °C for 5 min.

Purification and size selection of the library was then performed using Agencourt AMPure XP beads (Beckman Coulter, Brea, CA, USA), first using 0.5X bead:DNA ratio to remove long and secondly 1.8X to remove short (i.e., adapter dimers) fragments, respectively. Library concentration was measured with a high-sensitivity DNAchip on a Bioanalyzer 2100 (Agilent, Santa Clara, CA, USA). Finally, the library was sequenced on an Illumina NovaSeq6000 SP lane with a 2 × 50 bp setup, incl. Xp kit (validated method) at the *SciLifeLab* sequencing facility in Stockholm.

Extractions and library preparation were conducted in a separate aDNA lab and appropriate precautions were taken to minimize the risk of contamination of the ancient sample[36].

**Data processing, genome reconstruction and analyses**. First, raw aDNA data was demultiplexed using bcl2Fastq v2.17.1 with default settings (Illumina Inc.). We then trimmed adapters and merged paired-end reads using SeqPrep v1.1[37] with default settings but with a minor modification in the source code, allowing us to choose the best quality scores of bases in the merged region instead of aggregating the scores following Palkopoulou et al.[11]. We then merged sequencing reads and mapped them against the reference mitogenomes for chicken (*Gallus gallus*; GenBank: NC_001323) mitogenome using the BWA v0.7.13 aln algorithm[38] and slightly modified default settings with deactivated seeding (−l 16,500), allowing more substitutions (−n 0.01) and allowing up to two gaps (−o 2). The BWA samse command was then used to generate alignments in SAM format. The resulting reads were then processed in SAMtools v1.9[39], converted to BAM format coordinate sorted and indexed. We removed duplicates from the alignments using a custom python script to avoid inflation of length distribution for loci with deep coverage[11]. Next, we used Picard v1.141 (http://broadinstitute.github.io/picard) to assign read group information including library, lane and sample identity to each bam file. Reads were then re-aligned around indels using GATK v3.4.0[40]. Only reads/alignments with mapping quality ≥30 were kept for subsequent analysis. We then estimated the endogenous mitochondrial DNA content (i.e., the proportion of reads mapping to the mitogenome) and coverage with SAMtools.

Secondly, we reconstructed a consensus mitogenome using a majority consensus rule and a minimum of 5X coverage in Geneious® v7.0.336[41] and the raw data mapped to chicken. We extracted a partial COI gene (232 bp) from this mitogenome and used it as input for species identification in BLASTn[13,14].

Third, after identifying a 100% identity match with the horned lark (*Eremophila alpestris*), we generated a de novo mitogenome for a modern horned lark (*E. a. flava*), using whole genome sequencing data from Sigeman et al.[15], with Mitobim v1.9.0[16] using the 'quick' option and the Mongolian lark (*Melanocorypha mongolica;* GENBANK: NC_036760), which is the closest-related species with a compete mitogenome[17] as bait reference. Reconstruction of this mitogenome was done after four iterations. Next, we remapped the raw data for the ancient horned lark to the modern lark mitogenome, built its consensus genome in Geneious®

using the same parameters as above and estimated the endogenous mitochondrial DNA content, as described above. We also extracted the *cytb* (987 bp) and COI (683 bp) genes to confirm species identity in BLASTn[13,14].

Fourth, we investigated the phylogenetic relationships between the ancient horned lark and modern horned larks by including it in a mitochondrial data set consisting of 2034 bp (996 bp cytochrome *b* and 1038 bp ND2) from Ghorbani et al.[20]. We only included specimens without missing data, making for a total of 31 modern larks. The evolutionary relationship was inferred by implementing a Bayesian relaxed clock model in BEAST2 v2.4.0[21]. We ran Markov chain Monte Carlo chains for 80 million generations (sampling every 100 generations) using a relaxed lognormal distribution for the molecular clock model and assuming a birth-death speciation process for the tree prior. We used a GTR + G and HKY + G model for *cytb* and for ND2, respectively based on Ghorbani et al.[20]. The tree was calibrated by using the mean radio carbon date value for ancient horned lark as tip-date. We checked for convergence between runs and analysis performance using Tracer v1.6[42] and accepted the results if the values of the estimated sample size (ESS) were larger than 200, suggesting little autocorrelation between samples. The resulting trees were combined in TreeAnnotator v1.7.5[43] and the consensus tree with the divergence dates was visualized in FigTree v1.4.3[44].

Fifth, to estimate the overall endogenous DNA content, we mapped all raw data to the complete nuclear genome of the modern horned lark (Sigeman et al.[15]; N50 = 20 kb, 243,805 contigs) using the same mapping parameters as described above for mapping data to mitogenomes.

Finally, to determine the sex of the bird, we counted the number of reads mapping to autosomal (0.95 Gb) and Z-linked contigs (60.7Mb[15]) and calculated the average number of reads mapping to each contig groups per block of 100 kb. In birds, males are the homogametic sex (ZZ) while females are the heterogametic sex (ZW). Thus, a male should have the same proportion of reads mapping to autosomal and Z-linked contigs, making for a Z/autosomal ratio of ~1. Conversely, a female should have half the number of reads mapping to the Z-link contigs compared to autosomal contigs, making for a Z/autosomal ratio of ~0.5. A total of 22,356 and 735,448 reads mapped to the to Z-linked ($n = 2006$) and autosomal ($n = 241,800$) contigs, respectively. This corresponded to an average of 37 and 78 reads per 100 kb mapped to Z-linked and autosomal contigs, respectively.

**Reporting summary**. Further information on research design is available in the Nature Research Reporting Summary linked to this article.

## Data availability
Raw fastq files and bam files mapped to the de novo horned lark mitogenome are deposited at the European Nucleotide Archive (ENA; Proj ID PRJEB35255). Concatenated cytb+ND2 sequences are provided in Supplementary Data 1.

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

## Acknowledgements

Genome sequencing and radiocarbon dating was funded through a grant from FORMAS (grant no. 2018-01640). The authors acknowledge Boris Berezhnov and Spartak Khabrov for access to the bird remains. The authors also acknowledge support from the Uppsala Multidisciplinary Centre for Advanced Computational Science for assistance with massively parallel sequencing and access to the UPPMAX computational infrastructure. Sequencing was performed by the Swedish National Genomics Infrastructure (NGI) at the Science for Life Laboratory, which is supported by the Swedish Research Council and the Knut and Alice Wallenberg Foundation. The field work was supported by Renegade Pictures UK Ltd. Open access funding provided by Stockholm University.

## Author contributions

N.D. and L.D. conceived the study. D.W.G.S. performed DNA extractions and library preparation. N.D. and P.G.P.E. analysed the data. N.D. wrote the manuscript. N.D., D.W.G.S., H.S., P.G.P.E., J.G., D.C.F., A.V.P., V.L.H., V.P., B.H., and L.D. contributed to the final version of the manuscript and approved it for publication.

## Competing interests

The authors declare no competing interests.
