## [Peer Review File · Communications Biology]

REVIEWERS' COMMENTS:

Reviewer #1 (Remarks to the Author):

This work is a good example of the use of ancient DNA to understand the past populations on non-mammal species. The authors are very clear in how aDNA can be used to identify species, explore phylogenetic relationships, as well as to genetically identify specimens' sex. I find it an interesting work for publication, and I only have some minor comments.

General comment. The use of the Oxford comma is not consistent throughout the text. For instance, it is used in line 42 ("understanding of the impact of climate change on species, populations, and communities), but not in line 40 ("woolly rhinoceroses, horses, bison and wolverine). I would suggest to choose one form and keep it consistent.

Minor comments and suggestions:

1) Line 68, 158, and 161. Add a space before "mg".

2) Lines 169 – 206. Referencing to companies is not consistent in the methods section. For instance, the authors indicate "T4 DNA ligase (Thermo Scientific)". However, there is no company reference for Bst Polymerase or MinElute. I would suggest to revise this section and refer to all the companies consistently.

3) Lines 224 and 236. It is a bit confusing to use the expression "endogenous DNA content" when only referring to the mitochondrial reads mapping to the reference, as in general is used to indicate the overall (nuclear and mitochondrial) proportion of reads mapping to the reference. I would rephrase it to "endogenous mitochondrial DNA content".

4) Lines 226 – 227 – To call the consensus sequence using Geneious, did you specify any minimum coverage requirements per base (e.g. at least 3 bases to call that position)? If, so, indicate.

5) Line 232 – Remove double parenthesis in "GENBANK: NC_036760))"

6) Line 236. Was the consensus genome also built using the majority consensus rule? What about the coverage requirements?

7) Line 239 – 240. Did you use a software like PartitionFinder to evaluate the need of partitioning the data for the BEAST2 analysis?

8) Line 245 – In the BEAST2 analysis, you used a gamma substitution model. Why did you select that model? How did you test that it was the best model for your data?

Reviewer #2 (Remarks to the Author):

This is an excellent, concise manuscript that will be of great interest to a broad array of scientists. The methods are appropriate, timely, and meticulous. All conclusions are justified by the data presented. I have no particular suggestions to improve this manuscript, which I recommend highly for publication.

David Steadman

Reviewer #3 (Remarks to the Author):

Dusseux et al. (submitted) provide the first documented evidence of avian carcass survivorship in permafrost, dated to ca. 44,163–48,752 BP. The bird, a small passerine, was identified as a horned lark (*Eremophila alpestris*) by comparing a partial COI gene (232 bp) to the GenBank avian genetic database. It was assigned female based on Z-linked/autosomal readings (46%) which implied the presence of a W chromosome. The subsequent data on lark phylogeny, environment, and climate change that this specimen provides justifies its significance for publication in *Communications Biology*.

Eremophila is a widespread, morphologically variable taxon. Using molecular analysis of mitochondrial cytochrome-b and ND2 Ghorbani et al. (2019) had predicted an initial divergence of *Eremophila* sometime during the Plio-Pleistocene transition, with origination of the horned lark, *Eremophila alpestris* (*E. a. alpestris*, *E. a. flava*, and *E. a. brandti*) during the middle Pleistocene. This permafrost specimen's phylogenetic placement near the node for *E. a. flava* (northern Palearctic tundra) and *E. a. brandti* (central Palearctic steppe) places it as an ancestral species with subsequent sub-speciation of *E. alpestris* occurring sometime after 45,000 years ago. This suggests a Last Glacial Maximum sub-speciation event that occurred as mixed tundra steppe shifted into separate tundra (northern) and steppe (central) environs. Oftentimes, genetic data is used to better understand ancient lineages and to predict points of divergence in the fossil record; this is an example of how paleontological remains can, if exceptionally preserved, not only support but refine these data. This supports continued exploration of permafrost environments for other exceptionally preserved faunal remains.

This paper documents climatic fluctuations and subsequent vegetative changes leading to speciation of *Eremophila alpestris* and provides insight on how climate change effects species. It is therefore of great significance to both the fields of conservation and paleontology and to the study of biodiversity and climate change.

Overall, this manuscript is well-written and organized and provides the necessary data and supportive text to find this study convincing and its findings significant. Understanding this is a first submission, I would like to see better figures. Figure 1a is dark and the scale unclear and I am not sure why there is a supplemental close-up image of the specimen's leg, unless this was to provide further evidence of its preservation state. I would prefer to see the carcass images as a single Figure 2 with Figure 1 providing a map and perhaps a photograph of the site, as I imagine most readers will be unfamiliar with this type of excavation site.

The recovery of a complete passerine carcass in permafrost is, in itself, fascinating and I wonder if other smaller, non-avian animals have been recovered previously, or if past finds have been limited to larger mammals? Line 119 notes other permafrost faunal specimens from these deposits; is there evidence that fossil ivory collectors may have damaged other such specimens and do they have an estimated time-line of when the initial excavation tunnels were made compared to when the specimen was collected?

I am also curious to know the condition of the specimen's skeleton, as it would have been subjected to numerous freeze-thaw cycles, and wonder if a CT scan was attempted. This could also potentially provide seasonal information if medullary bone were found to be present.

While I am not the best person to attest to the methodologies used for mitogenomes reconstruction, the authors provide these in a clear and concise way, with adequate references, to justify the methods

and allows a non-specialist to understand the methods. This is important for the broader scientific audience of Communications Biology.

REVIEWERS' COMMENTS:

Reviewer #1 (Remarks to the Author):

This work is a good example of the use of ancient DNA to understand the past populations on non-mammal species. The authors are very clear in how aDNA can be used to identify species, explore phylogenetic relationships, as well as to genetically identify specimens' sex. I find it an interesting work for publication, and I only have some minor comments.

General comment. The use of the Oxford comma is not consistent throughout the text. For instance, it is used in line 42 (“understanding of the impact of climate change on species, populations, and communities), but not in line 40 (“woolly rhinoceroses, horses, bison and wolverine). I would suggest to choose one form and keep it consistent.

>>> *we have added a comma on l. 40, l. 177, l. 202, l. 209*

Minor comments and suggestions:

1) Line 68, 158, and 161. Add a space before “mg”.

>>> *spaces have been added*

2) Lines 169 – 206. Referencing to companies is not consistent in the methods section. For instance, the authors indicate “T4 DNA ligase (Thermo Scientific)”. However, there is no company reference for Bst Polymerase or MinElute. I would suggest to revise this section and refer to all the companies consistently.

>>> *Companies have been added on l. 178 and 195*

3) Lines 224 and 236. It is a bit confusing to use the expression “endogenous DNA content” when only referring to the mitochondrial reads mapping to the reference, as in general is used to indicate the overall (nuclear and mitochondrial) proportion of reads mapping to the reference. I would rephrase it to “endogenous mitochondrial DNA content”.

>>> *The word ‘mitochondrial’ has added been on l. 231 and 244*

4) Lines 226 – 227 – To call the consensus sequence using Geneious, did you specify any minimum coverage requirements per base (e.g. at least 3 bases to call that position)? If, so, indicate.

>>> *We used a minimum of 5X and added this information on l. 234*

5) Line 232 – Remove double parenthesis in “GENBANK: NC_036760))”

>>> *This has been removed on l. 240*

6) Line 236. Was the consensus genome also built using the majority consensus rule? What about the coverage requirements?

>>> *This information has been added on l. 244*

7) Line 239 – 240. Did you use a software like PartitionFinder to evaluate the need of partitioning the data for the BEAST2 analysis?

>>> *We used the PartitionFinder results from Ghorbani et al. (2019). We have now clarified this on l. 254-255*

8) Line 245 – In the BEAST2 analysis, you used a gamma substitution model. Why did you select that model? How did you test that it was the best model for your data?
>>> *We used the same substitution models as identified in Ghorbani et al. (2019). We have now clarified this on l. 254-255*

Reviewer #2 (Remarks to the Author):

This is an excellent, concise manuscript that will be of great interest to a broad array of scientists. The methods are appropriate, timely, and meticulous. All conclusions are justified by the data presented. I have no particular suggestions to improve this manuscript, which I recommend highly for publication. David Steadman
>>> *We are very happy to hear that Dr. Steadman enjoyed reviewing our manuscript and thank him for his positive recommendation.*

Reviewer #3 (Remarks to the Author):

Dussex et al. (submitted) provide the first documented evidence of avian carcass survivorship in permafrost, dated to ca. 44,163–48,752 BP. The bird, a small passerine, was identified as a horned lark (*Eremophila alpestris*) by comparing a partial COI gene (232 bp) to the GenBank avian genetic database. It was assigned female based on Z-linked/autosomal readings (46%) which implied the presence of a W chromosome. The subsequent data on lark phylogeny, environment, and climate change that this specimen provides justifies its significance for publication in *Communications Biology*.

Eremophila is a widespread, morphologically variable taxon. Using molecular analysis of mitochondrial cytochrome-b and ND2 Ghorbani et al. (2019) had predicted an initial divergence of *Eremophila* sometime during the Plio-Pleistocene transition, with origination of the horned lark, *Eremophila alpestris* (*E. a. alpestris*, *E. a. flava*, and *E. a. brandti*) during the middle Pleistocene. This permafrost specimen's phylogenetic placement near the node for *E. a. flava* (northern Palearctic tundra) and *E. a. brandti* (central Palearctic steppe) places it as an ancestral species with subsequent sub-speciation of *E. alpestris* occurring sometime after 45,000 years ago. This suggests a Last Glacial Maximum sub-speciation event that occurred as mixed tundra steppe shifted into separate tundra (northern) and steppe (central) environs. Oftentimes, genetic data is used to better understand ancient lineages and to predict points of divergence in the fossil record; this is an example of how paleontological remains can, if exceptionally preserved, not only support but refine these data. This supports continued exploration of permafrost environments for other exceptionally preserved faunal remains.

This paper documents climatic fluctuations and subsequent vegetative changes leading to speciation of *Eremophila alpestris* and provides insight on how climate change effects species. It is therefore of great significance to both the fields of conservation and paleontology and to the study of biodiversity and climate change.

Overall, this manuscript is well-written and organized and provides the necessary data and supportive text to find this study convincing and its findings significant. Understanding this is a first submission, I would like to see better figures. Figure 1a is dark and the scale unclear and I am not sure why there is a supplemental close-up image of the specimen's leg, unless

this was to provide further evidence of its preservation state. I would prefer to see the carcass images as a single Figure 2 with Figure 1 providing a map and perhaps a photograph of the site, as I imagine most readers will be unfamiliar with this type of excavation site.

>>> *We now provide higher-resolution figures in the resubmission. We have however kept Figure S1(now Figure 4) as it can be used for species determination and also to further document the state of preservation of the specimen. We also separate Fig. 1 and 2 as suggested by reviewer #3 and only show one photo with ventral view of the specimen.*

The recovery of a complete passerine carcass in permafrost is, in itself, fascinating and I wonder if other smaller, non-avian animals have been recovered previously, or if past finds have been limited to larger mammals? Line 119 notes other permafrost faunal specimens from these deposits; is there evidence that fossil ivory collectors may have damaged other such specimens and do they have an estimated time-line of when the initial excavation tunnels were made compared to when the specimen was collected?

>>> *Damage to permafrost specimens by fossil ivory collectors are certainly possible. However, without these excavations, the discovery of such specimens would be much more rare due to that most naturally eroded specimens would not be discovered before either rotting away or being consumed by scavengers. We have added a statement about this issue on l. 120-122. While we do not have exact information on the timeline of the excavation, it is our understanding that the tunnel where the bird was found has been actively excavated for at least two years.*

I am also curious to know the condition of the specimen's skeleton, as it would have been subjected to numerous freeze-thaw cycles, and wonder if a CT scan was attempted. This could also potentially provide seasonal information if medullary bone were found to be present.

>>> *We think it is highly unlikely that the specimen has been subjected to multiple freeze-thaw cycles. This bird has been buried many meters down in the permafrost, and has thus likely been constantly frozen since its death. We thank the reviewer for suggesting a CT scan to try and see if there is a medullary bone. This is a great idea for a future project. At present, the specimen is stored in Yakutsk where high-resolution CT scanners are not available, but there are discussions to send the specimen on a temporary loan to Sweden in the future, possibly next year, and hopefully a CT-scan can be arranged at that date.*

While I am not the best person to attest to the methodologies used for mitogenomes reconstruction, the authors provide these in a clear and concise way, with adequate references, to justify the methods and allows a non-specialist to understand the methods. This is important for the broader scientific audience of Communications Biology.

>>> *We are glad to read that our methods are clear even for a non-specialist readership.*